# ERAP2 Inhibition Induces Cell-Surface Presentation by MOLT-4 Leukemia Cancer Cells of Many Novel and Potentially Antigenic Peptides

**DOI:** 10.3390/ijms23031913

**Published:** 2022-02-08

**Authors:** Ioannis Temponeras, George Stamatakis, Martina Samiotaki, Dimitris Georgiadis, Harris Pratsinis, George Panayotou, Efstratios Stratikos

**Affiliations:** 1National Centre for Scientific Research “Demokritos”, 15341 Agia Paraskevi, Greece; john.t13111@windowslive.com (I.T.); hprats@bio.demokritos.gr (H.P.); 2Department of Pharmacy, University of Patras, 26504 Patra, Greece; 3Biomedical Sciences Research Center “Alexander Fleming”, Institute for Bioinnovation, 16672 Vari, Greece; stamatakis@fleming.gr (G.S.); samiotaki@fleming.gr (M.S.); g.panayotou@fleming.gr (G.P.); 4Department of Chemistry, National and Kapodistrian University of Athens, 15784 Zografou, Greece; dgeorgia@chem.uoa.gr

**Keywords:** aminopeptidase, antigenic peptide, antigen presentation, adaptive immunity, major histocompatibility molecules, proteomics, immunopeptidome, inhibitor

## Abstract

Recent studies have linked the activity of ER aminopeptidase 2 (ERAP2) to increased efficacy of immune-checkpoint inhibitor cancer immunotherapy, suggesting that pharmacological inhibition of ERAP2 could have important therapeutic implications. To explore the effects of ERAP2 inhibition on the immunopeptidome of cancer cells, we treated MOLT-4 T lymphoblast leukemia cells with a recently developed selective ERAP2 inhibitor, isolated Major Histocompatibility class I molecules (MHCI), and sequenced bound peptides by liquid chromatography tandem mass spectrometry. Inhibitor treatment induced significant shifts on the immunopeptidome so that more than 20% of detected peptides were either novel or significantly upregulated. Most of the inhibitor-induced peptides were 9mers and had sequence motifs and predicted affinity consistent with being optimal ligands for at least one of the MHCI alleles carried by MOLT-4 cells. Such inhibitor-induced peptides could serve as triggers for novel cytotoxic responses against cancer cells and synergize with the therapeutic effect of immune-checkpoint inhibitors.

## 1. Introduction

Cancer actively tries to evade the human immune response through the process of immunoediting [1]. Thus, clinical interventions that re-program the immune system to better detect and eradicate cancer can be invaluable assets for the treatment of the disease. In such an approach, the use of immune-checkpoint inhibitors (ICI) in cancer immunotherapy has demonstrated impressive successes that are changing the landscape of cancer treatment [2]. Unfortunately, the clinical success of ICI therapies is often limited to a subset of patients [3], and recent efforts have focused on determining the causes for such resistance [4]. Recently, the antigen presentation pathway has emerged as a key regulator of response to ICI therapies [5]. 

Cytotoxic T-lymphocytes can kill cancerous cells after recognizing small peptides derived from cancer-specific antigens. These small peptides, called antigenic peptides if they elicit an immune response, are presented as a complex with major histocompatibility complex class I molecules (MHCI, also called human leukocyte antigens, HLA, when referring to the human genes) on the cell surface and are produced by complex proteolytic cascades inside the cell [6]. The sum of the presented peptides is called the immunopeptidome and represents the proteome status of the cell [7]. Intracellular aminopeptidases such as endoplasmic reticulum aminopeptidase 1 (ERAP1) and endoplasmic reticulum aminopeptidase 2 (ERAP2) play important roles in the generation of antigenic peptides and indirectly regulate adaptive immune responses [8]. They achieve this by either trimming elongated precursors down to the optimal length for MHCI binding (usually nine amino acids long), or by over-trimming antigenic peptides to lengths too short for MHCI binding, effectively destroying them. Both ERAP2 and its homologous ERAP1 have been shown to regulate the cellular immunopeptidome [9]. In particular, ERAP2 has been shown to underlie changes in peptidomes of particular HLA alleles related to predisposition to inflammatory autoimmunity, [10,11,12,13], although its role on regulating the global immunopeptidome has not been thoroughly studied. Accordingly, high ERAP2 expression levels have been associated with predisposition to inflammatory autoimmunity [14], suggesting that antigen processing by ERAP2 could be a key regulator of cellular antigenicity in human disease [15]. This was recently demonstrated for the homologous enzyme, ERAP1, in the case of Psoriasis, since reduced ERAP1 activity resulted in the reduced generation of an autoantigenic peptide that sustains cytotoxic T cell activity against melanocytes in patients [16]. ERAP1, being the dominant aminopeptidase that generates antigenic peptides, has already been recognized as a potential therapeutic target in cancer immunotherapy [17,18]. However, recent results have highlighted that ERAP2 may have an even greater impact on the efficacy of ICI immunotherapy. Impressively, reduced ERAP2 expression levels due to common genetic polymorphisms have been recently shown to predict survival in patients treated with ICI immunotherapy in 24 out of 24 cancer types examined [19]. Accordingly, ERAP2 has been shown to be upregulated in several cancers [20]. Therefore, it is possible that pharmacological inhibition of ERAP2 could enhance the efficacy of cancer immunotherapy and extend it to more patients. 

Inspired by these findings, several studies have explored the development of ERAP2 inhibitors, although their effects on cellular models is currently lacking [21]. In this study, we utilized compound DG011A (described as compound **6g** in [22]), to our knowledge the most potent and selective inhibitor of ERAP2 described, to explore changes in the immunopeptidome of the MOLT-4 T lymphoblast leukemia cell line. Although inhibitor treatment had only a minor effect in MHCI expression on the cell surface, it induced a significant immunopeptidome shift, leading to the presentation of many new peptides that have the potential to be immunogenic. Our work suggests that, in cancer cells, ERAP2 plays important roles in limiting presentation of cancer antigens and thus pharmacological inhibition by DG011A or similar inhibitors may be a viable approach to modulating the cancer immunopeptidome for immunotherapy applications.

## 2. Results and Discussion

Phosphinic pseudopeptides can act as transition state analogues and are potent inhibitors of several members of the oxytocinase sub-family of M1 aminopeptidases. During a previous SAR study, compound DG011A (described as compound **6g** in [22]) was identified to be a potent inhibitor of ERAP2 with good selectivity versus the homologous ERAP1 [22]. Indeed, DG011A inhibits the trimming of the fluorigenic substrate Arg-AMC by ERAP2 with an IC_50_ = 89 nM, while it inhibits the trimming of the fluorigenic substrate Leu-AMC by ERAP1 with an IC_50_ = 6.4 μM, a 72-fold difference (Figure 1A). To evaluate the effect of this compound on the immunopeptidome of cancer cells, we selected the MOLT-4 T lymphoblast leukemia cell line because it has a good basal level of expression of ERAP2 (Appendix A). In addition, expression levels of ERAP1 are much lower than ERAP2 in MOLT-4 cells, making ERAP2 the dominant aminopeptidase in the ER and thus facilitating the observation of changes in the immunopeptidome after ERAP2 inhibition (Appendix A). At the same time, co-expression of ERAP1 and ERAP2 in MOLT-4 cells (both enzymes are expressed in most cancer cell lines [20]) allows us to discern the effects of ERAP2 in the immunopeptidome in more native conditions in which ERAP1 is also a contributing factor. DG011A demonstrated no apparent toxicity on MOLT-4 cells as evidenced by the MTT assay, for concentrations up to 100 μM (Figure 1B). Treating MOLT-4 cells with 1 μΜ DG011A for 48 h resulted in only a small decrease in cell-surface staining of MHCI, suggesting that the inhibitor does not abrogate overall antigen presentation by the cell (Figure 1C,D). 

To analyze the effects of the compound on the immunopeptidome of MOLT-4 cells, the cells were cultured in the presence of 1 μM compound, grown to 0.5 × 10^9^ cells, harvested and the MHCI-peptide complexes isolated by affinity chromatography using the pan-HLA monoclonal antibody W6/32 as previously described [23]. Eluted peptides from the MHCI complexes were sequenced by LC-MS/MS. Overall, we performed three biological replicates of each condition (with and without the inhibitor) and each was analyzed in three technical replicates, bringing the total to nine replicates for each condition. The unfiltered lists of identified peptides, along with relevant identification parameters, are shown in Appendix A. A scatterplot of identified peptides comparing the relative signal intensity between the two conditions is shown in Figure 2A. Most peptides lie in the diagonal, suggesting that they are detected in similar amounts irrespective of the presence of the inhibitor. Still, a significant number of peptides lie outside the diagonal, indicating that they are either up- or down-regulated by the inhibitor. Several peptides were also detected uniquely in a single condition (indicated on the edges of the plot near each axis). A heatplot of all measured signals is shown in Figure 2B. The heatplot suggests a good degree of reproducibility between replicates. Interestingly, a cluster of peptides, upregulated by the inhibitor, is evident (in the top right section of the plot), suggesting that the presence of the inhibitor induced the presentation of many peptides. To take advantage of our replicate measurements to enhance the statistical robustness of the comparison, we limited further comparisons to peptides that were detected to change with a *p* value < 0.05 as shown in the Volcano plot in Figure 2C. Overall, we identified 1394 peptides of which 1040 were unchanged by the inhibitor, 72 were down-regulated by more than five-fold and 282 were novel or upregulated by more than 5-fold (Figure 2D). These results suggested that the inhibitor had a significant effect on antigen presentation and induced or enhanced the presentation of many peptides on the cell-surface. This result is overall consistent with the proposed role of ERAP2 in complementing ERAP1 in shaping the immunopeptidome, primarily by destroying some antigenic epitopes [15].

Peptide length is a very important parameter for binding onto MHCI due to the size restrictions of the peptide binding groove; most peptides presented by MHCI are 11 residues or shorter [24]. To examine if the ERAP2 inhibitor had any effect on the length of presented peptides we plotted the distribution of peptide lengths for the peptides that were unaffected by the inhibitor and the peptides that were induced by the inhibitor (Figure 3A,B and Appendix A). Peptides that were not affected by the inhibitor were predominantly 9mers, as expected based on the length preferences of MHCI alleles, although a small number of longer peptides were also detectable (Figure 3A). Similarly, peptides that were up-regulated by the inhibitor also were predominantly 9mers, but included a few additional longer peptides. Still, there was no clear shift in peptide length, as seen previously when using an ERAP1 inhibitor [23]. This observation is consistent with the known molecular mechanisms of these two enzymes, since ERAP1 has been shown to trim many peptides based on their length by using a distinct regulatory site [25,26], whereas ERAP2 does not appear to share the same property [27]. Rather, the presence of some additional elongated longer peptides induced by the inhibitor is likely the result of a reduced aminopeptidase activity in the ER.

To validate that the detected peptides are indeed HLA ligands and presented by one of the HLA-alleles present in MOLT-4 cells [28], we utilized the HLAthena prediction server [29] to rank the peptides with a length of 8–12 residues (based on the length limitations of the prediction server) for binding to one of the alleles *HLA-A*01:01, HLA-A*25:01, HLA-B*18:01, HLA-B*57:01, HLA-C*06:02, HLA-C*12:03* (Figure 3C). Peptides marked with a rank below 2.0 are considered HLA-binders (cyan region in Figure 3C). More than 80% of the peptides identified to be common in both conditions were predicted to be binders for at least one of the HLA alleles in MOLT-4 cells, whereas more than 75% of the peptides induced by the inhibitor were also predicted to be binders. This result both validates our isolation and detection protocols, but also suggests that the vast majority of the peptides that are up-regulated by the inhibitor are indeed presented by MHCI and therefore have the capacity to become immunogenic. To further validate this result, we analyzed the sequence of the 9mer peptides that were common in both the control and inhibitor conditions or were induced by the inhibitor, using the GibbsCluster-2.0 Server [30] in order to reveal sequence patterns. Two major clusters were evident from the analysis and are depicted in Figure 4. Comparison of these clusters to known peptides that bind to one of the MHCI alleles expressed by MOLT-4 cells (extracted from http://hlathena.tools/ (accessed on 27 December 2021) [29], Appendix A) indicated that the two clusters found for the common (unaffected) peptides likely represent peptide presentation by *HLA-A*25:01, HLA-B*18:01* and *HLA-C*18:03* (Figure 4A) and *HLA-A*01:01* and possibly *HLA-B*57:01* (Figure 4B) respectively. Similarly, the two main clusters found for peptides that were induced by the inhibitor, likely represent presentation primarily by *HLA-A*25:01* (Figure 4C) and *HLA-A*01:01* (Figure 4D) respectively, although peptides presented by *HLA-B*18:01* and *HLA-C*06:02* is also likely. Interestingly, inhibitor-induced peptides appear to be more biased towards *HLA-A*25:01*, hinting that this allele may tend to present peptides that are more sensitive to over-trimming by ERAP2, which are spared in the presence of the inhibitor.

In summary, we present evidence that a recently developed ERAP2 inhibitor, with good selectivity versus ERAP1, has the capability to regulate the global immunopeptidome of a cancer cell line and to induce the presentation of many new peptides that are good MHCI ligands and therefore have the capacity to be immunogenic when presented in an immunocompetent host. Our results accentuate one of the proposed biological roles of ERAP2, specifically its ability to destroy antigenic peptides and thus limit presentation of some cancer antigens. Based on our findings, we propose that chemical inhibition of ERAP2 activity can spare the destruction of MHCI ligands that can then become antigenic and enhance the immunogenicity of cancer cells. Our study, however, also carries some important limitations: while the observed results are consistent with ERAP2 inhibition, unknown off-target effects of the inhibitor could also, indirectly, exert effects on the immunopeptidome. Furthermore, the generality of the observed effects on other cancer cells is currently unknown and could heavily depend on the role of ERAP2 antigenic peptide trimming in the cells examined and as a result could be limited to cells that express high levels of ERAP2. Lastly, our approach did not address the relative contribution of ERAP1 and ERAP2 in shaping the immunopeptidome in this system. Besides these limitations, we provide proof-of-concept of the importance of pharmacological manipulation of ERAP2 enzymatic activity on antigen presentation even in the presence of the homologous ERAP1.

The over-expression of ERAP2 in cancer is likely an immune-evasion mechanism and therefore the pharmacological restriction of ERAP2 activity could enhance the antigenicity of cancer cells and possibly synergize with checkpoint-inhibitor cancer immunotherapy. While the generality and clinical relevance of such an approach is currently unknown, further pharmacological development of DG011A or similar inhibitors may constitute a worthwhile effort that can generate useful chemical tools and drug leads towards the long-term goal of improving the efficacy of cancer immunotherapy. 

## 3. Materials and Method

### 3.1. Cell Culture

MOLT-4 cells (obtained from ATCC, Manassas, Virginia, cat. no. CRL-1582) were cultured in RPMI 1640 supplemented with 2 mM glutamine, 10% heat-inactivated FBS (Gibco, Waltham, MA, USA), penicillin and streptomycin and incubated at 37 °C, 5% CO_2_. Cells were counted before each passage, to maintain a density between 4 × 10^5^–2 × 10^6^ cells/mL. Cell viability was monitored using the Trypan Blue dye during each passage, and was consistently over 95%.

### 3.2. Antibodies

For the immune-purification of the MHC-I molecules carrying the MOLT-4 immunopeptidome, the W6/32 monoclonal antibody was used. The antibody was isolated from the supernatant of hybridoma cells grown in culture and purified using protein G affinity chromatography. For FACS analysis, MHC-I molecules were stained with the W6/32 monoclonal antibody conjugated with FITC (Biorad, Hercules, CA, USA, MCA81F). For the detection of ERAP1 and ERAP2 in cell lysates by western blot the following primary antibodies were used: human aminopeptidase PILS/ARTS1 polyclonal goat IgG (R&D Systems, Minneapolis, MN, USA, AF2334) for ERAP1 and human aminopeptidase LRAP/ERAP2 polyclonal goat IgG (R&D Systems, Minneapolis, MN, USA, AF3830) for ERAP2. As a secondary antibody anti-goat IgG–HRP (HAF017) was also purchased from R&D systems.

### 3.3. Recombinant Proteins and Enzymatic Assays

Recombinant ERAP1 and ERAP2 were produced from baculovirus infected insect cells (Hi5™, Thermo Fischer, Waltham, MA, USA) as described previously [31]. Enzymatic titrations to evaluate the in vitro efficacy of the inhibitor were performed using a small fluorescent substrate assay as described previously [27,31]. 

### 3.4. Western Blotting

About 10 × 10^6^ MOLT-4 cells were lysed with 500 μL lysis buffer containing 1% Triton X-100, 0.1% sodium deoxycholate, 1 mM EDTA, pH 8.0, complete protease inhibitors (Roche, Basel, Switzerland: 12326400) in 50 mM Tris–HCl, pH 7.5, and 150 mM NaCl buffer. After cell lysis, the total protein concentration was determined using bicinchoninic acid (BCA) Protein Assay Kit (Thermo Scientific, Waltham, MA, USA). Whole cell lysates were analyzed with SDS-PAGE under reducing conditions in a 10% polyacrylamide gel. The separated proteins were blotted onto a polyvinylidene difluoride (PVDF) membrane (Thermo Scientific, Waltham, MA, USA) using the antibodies described above. Primary antibodies were used at a final concentration of 2 μg/mL and the secondary antibodies were diluted 1:1000. As an HRP substrate, we used the Pierce chemiluminescent western blotting substrate (Thermo Scientific, Waltham, MA, USA) and enhanced chemiluminescence was detected in LAS 4000 (Fujifilm, Greenwood, South Carolina). The images were processed using the AIDA Image analysis software 4.1 (Elysia-Raytest).

### 3.5. Synthesis of Phosphinic Inhibitor DG011

Phosphinic inhibitor DG011 [((1*R*)-1-amino-3-phenylpropyl){(2′*S*)-2′-[((2″*S*)-1″-amino-3″-hydroxy-1″-oxopropan-2″-yl)carbamoyl]-4′-methylpentyl} phosphinic acid] was prepared according to the synthetic strategy shown below (Figure 5). Briefly, phosphinic acid 1 reacted under silylating conditions with acrylate 2, the P-Michael product was saponified and diacid 3 was isolated after recrystallization of the resulting diastereoisomeric mixture, in 46% yield over 3 steps. Then, the Cbz protecting group was exchanged with Boc group, affording diacid 4 in 92% yield. DG011 was obtained in 43% overall yield after coupling of 5 with H-(L)Ser(TBS)-NH_2_ and acidic deprotection (See Appendix A). 

### 3.6. Cytotoxicity Assay

MOLT-4 cells (5000 cells/well) were cultured as described above in the presence of varying concentrations of DG011 (0–100 μM) for 48 h. The culture medium was replenished with 100 μL of RPMI 1640 containing 2 mg/mL MTT reagent, to a final concentration of 1 mg/mL. After incubation for another 4 h the cell cultures were centrifuged at 1250 rpm for 5 min at room temperature. The resultant formazan crystals were dissolved in 100 μL DMSO and the absorbance intensity was measured on a TECAN infinite M200 microplate fluorescence reader (Tecan Group Ltd., Männedorf, Switzerland) at 540 nm with reference at 620 nm. All experiments were performed at least three times and the relative cell viability (%) was expressed as a percentage relative to untreated control cells.

### 3.7. Treatment of MOLT-4 Cells with DG011 Inhibitor

MOLT-4 cells were treated with a DG011 inhibitor for a total of five days. The inhibitor was added to the complete culture medium at a final concentration 1 μM. The cell medium that contained the inhibitor was refreshed every two days. Cell cultures were incubated at 37 °C, 5% CO_2_. After five days of incubation, cells were harvested for immunopeptidome isolation or for flow cytometry. For immunopeptidome analysis, cells were cultured at a large scale and collected from flasks using Centrifuge 5430 R (Eppendorf, Hamburg, Germany). Cell pellets were stored at −80 °C until immunopeptidome isolation. For the flow cytometry analysis, cells were seeded in a 12-well plate and harvested with 10 mM EDTA pH 8.0, in PBS.

### 3.8. Flow Cytometry

Approximately 5 × 10^4^ cells per sample were transferred in FACS tubes and washed twice with 1 mL FACS buffer (1%BSA/PBS, 0.02% NaN_3_, 10 mM EDTA). The cells were stained with 4 μL undiluted W6/32 antibody labeled with Fluorescein (Biorad, Hercules, CA, USA, MCA81F) for 30 min on ice. After incubation, the cells were washed with 0.5 mL FACS buffer, centrifuged at 200× *g* for 5 min at 4 °C and re-suspended in 300 μL FACS buffer. The samples were analyzed in a FACScalibur flow cytometer using the BD CellQuest™ Pro software (Version 6.0, BD Bioschences, Franklin Lakes, NJ, USA). Approximately 20,000 events per sample were measured.

### 3.9. Preparation of Immunoaffinity Columns

W6/32 antibody (2 mg per column) was dialyzed in coupling buffer (NaHCO_3_ 0.1 M, NaCl 0.5 M, pH 8.3) overnight. To generate one 1 mL bed volume of cyanogen bromide-activated Sepharose 4B (GE Healthcare, Chicago, IL, USA, 17-0430-01), 0.285 g of dry beads was used. Sepharose was rehydrated with 1 mM HCl for 30 min and then washed thoroughly with a coupling buffer. The solution of the antibody was added to the beads and left for coupling overnight at 4 °C. After coupling, the beads were washed with coupling buffer and then with blocking buffer (Tris–HCl 0.1 M, pH 8.0). After the washes the beads were transferred to a 50-mL tube with blocking buffer and were mixed for 3 h at room temperature. Finally, the beads were washed with three cycles of acidic buffer (CH_3_COONa 0.1 M, NaCl 0.5 M, pH 4.0) and then basic buffer (Tris–HCl 0.1 M, NaCl 0.5 M, pH 8.0) solutions and then with 20 mM Tris–HCl, pH 7.5, 150 mM NaCl. For the pre-columns, the exact same procedure was followed except for the W6/32 coupling step. The columns and the pre-columns were stored at 4 °C until needed.

### 3.10. Isolation of MHC-I Immunopeptidome

For the isolation of the immunopeptidome, 0.5 × 10^9^ cells per sample were used. Cells were lysed with 20 mL lysis buffer (Tris–HCl, pH 7.5, 150 mM NaCl, 0.5% IGEPAL CA-630, 0.25% sodium deoxycholate, 1 mM EDTA pH 8.0, 1 × complete EDTA-free protease inhibitor cocktail tablets) for 1 h at 4 °C. The cell lysate was cleared with ultracentrifugation at 100,000× *g* for 1 h at 4 °C and then loaded onto a cyanogen bromide activated Sepharose pre-column, blocked as described above. The flow through from the pre-column was passed through W6/32-coupled beads three times and then washed with 20 bed volumes 20 mM Tris–HCl, pH 8.0, 150 mM NaCl, 20 bed volumes 20 mM Tris–HCl, pH 8.0, 400 mM NaCl, 20 bed volumes 20 mM Tris–HCl, pH 8.0, 150 mM NaCl and finally with 40 bed volumes 20 mM Tris–HCl, pH 8.0. The MHC-I–peptide complexes were eluted from the immunoaffinity column with 1% TFA. The peptides were separated from the MHC-I molecules using reversed-phase C18 disposable spin columns (Thermo Scientific, Waltham, MA, USA). The fraction containing peptides was dried prior to LC-MS/MS analysis.

### 3.11. LC-MS/MS Analysis 

The purified peptidic samples were pre-concentrated on a pepmap LC trapping column (Thermo Scientific, Waltham, MA, USA, 0.3 × 5 mm) at a rate of 5 μL of Buffer A (0.1% Formic acid in water) for 5 min. Then the samples were injected onto a 50 cm long pepmap column (Thermo Scientific, Waltham, MA, USA) placed in an oven set to 55 °C, using a gradient of 100% Buffer A (0.1% Formic acid in water) to 27.5% Buffer B (0.1% Formic acid in acetonitrile) in 35 min followed by an increase to 40% buffer B in 2.5 min and a second increase to 99% Buffer B in 0.5 min and then kept constant for 1 min. Finally, the column was equilibrated for 15 min prior to the subsequent injection. A full MS was acquired using a Q Exactive HF-X Hybrid Quadropole-Orbitrap mass spectrometer (Thermo Fischer, Waltham, MA, USA) in the scan range of 350–1500 m/z using 60 K resolving power with an AGC of 3 × 10^6^ and max IT of 45 ms, followed by MS/MS scans of the 12 most abundant ions, using 15K resolving power with an AGC of 1 × 10^5^ and max IT of 22 ms and an NCE of 28.

### 3.12. Data Analysis

The raw files were searched and the identified peptides were quantified in MaxQuant (https://www.maxquant.org/ accessed on 27 December 2021) (version 1.6.14.0, Martinsried, Germany), using “unspecific” search against the human uniprot protein database (downloaded 19/9/2019). Search parameters included a molecular weight ranging from 350 to 5000 Da, a precursor mass tolerance of 20 ppm, an MS/MS fragment tolerance of 0.5 Da and methionine oxidation, deamidation of asparagine and glutamine and protein N-terminal acetylation were set as variable modifications. The protein and peptide false discovery rate (FDR) was set to 1%. The match-between-run function was enabled. 

## Figures and Tables

**Figure 1 ijms-23-01913-f001:**
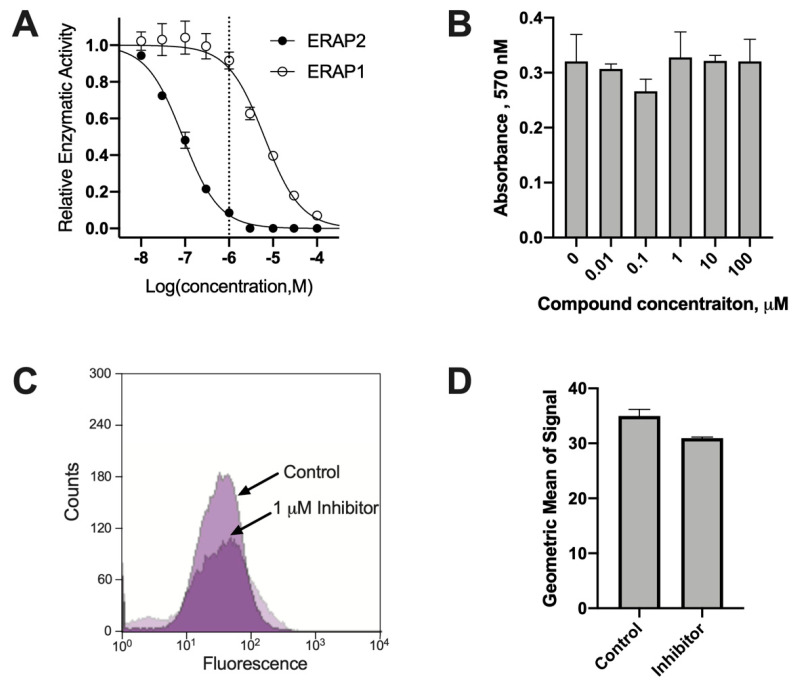
Activity of the phosphinic inhibitor DG011A. Panel (**A**), titration of DG011A inhibits ERAP2 with a 70-fold higher potency than ERAP1. Panel (**B**), DG011A shows no toxicity versus MOLT-4 cells up to 100 μM as measured by the MTT assay. Panel (**C**), representative FACS traces used for the quantitation of the presence of HLA molecules (stained by the W6/32 antibody) on the surface of MOLT-4 cells incubated with 1 μM DG011A. Panel (**D**), quantitation of the geometric mean of the signal from the FACS experiments.

**Figure 2 ijms-23-01913-f002:**
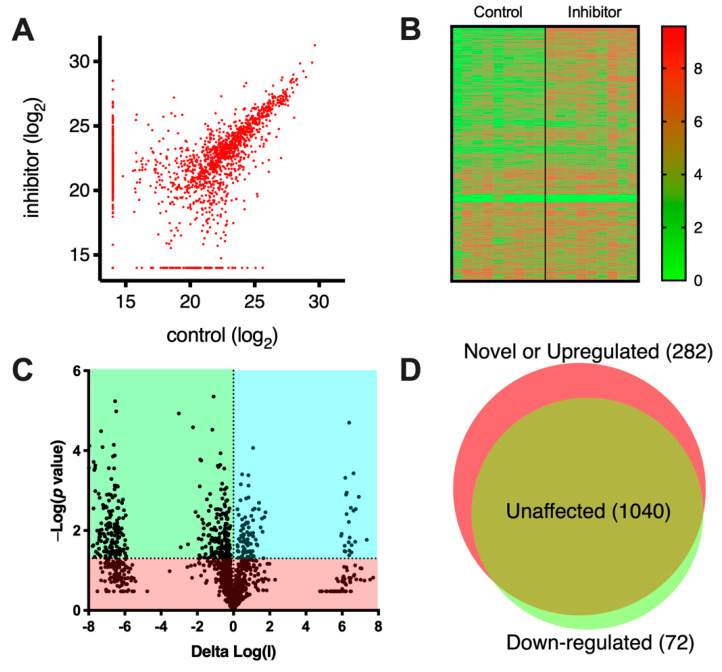
Effects of the DG011A on the immunopeptidome of MOLT-4 cells. Panel (**A**), scatterplot of the signal of detected peptides isolated from the HLA molecules of MOLT-4 cells under control conditions or after incubation with DG011A. Each circle represents a unique peptide sequence. Circles along the diagonal represent peptides unchanged between the two conditions and circles in the region close to each axis represent peptides detected only in a single condition (either control or inhibitor). Panel (**B**), heatplot showing the distribution of detected peptide signals (log_10_) in both conditions for each of the replicates measured (three biological replicates, each measured in three technical replicates, totaling 9 measurements per condition). Panel (**C**), volcano plot, indicating the statistical significance of the observed differences between the two conditions. Each circle represents a unique peptide sequence. The middle section represents peptides detected in both conditions but at different intensities and the outermost sections peptides detected in only a single condition. Peptides that fall within the green- and cyan-colored regions have a *p* value of <0.05 and are considered statistically significant. Panel (**D**), Venn diagram summarizing the observed numerical shifts of the immunopeptidome of MOLT-4 cells after incubation with DG011A.

**Figure 3 ijms-23-01913-f003:**
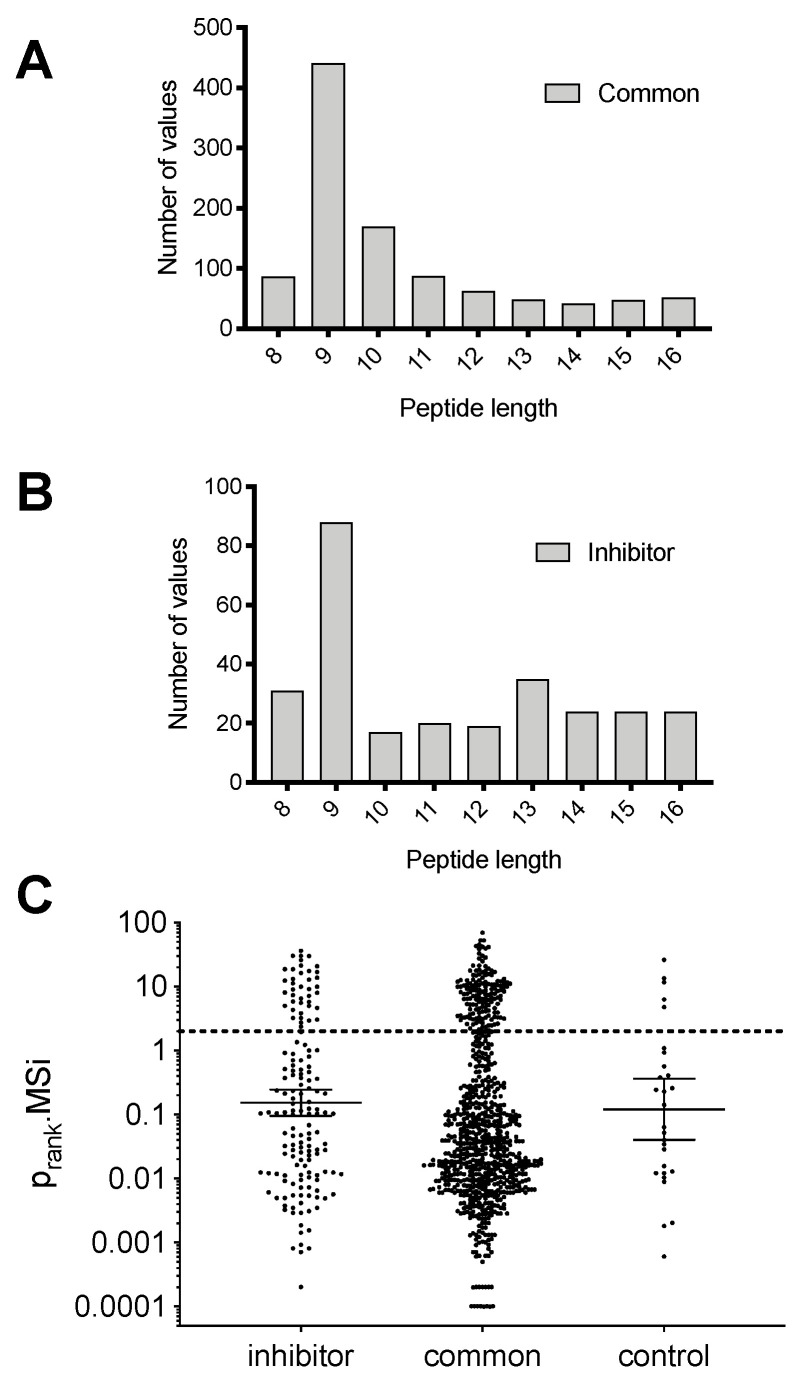
Effects of the inhibitor DG011A on the length and affinity of peptides presented by MOLT-4 cells. Panel (**A**), distribution of lengths of peptide eluted from the MHC class I molecules on the surface of MOLT-4 cells that are unaffected by the presence of the inhibitor. Panel (**B**), same as in panel A but for peptides that were induced by the inhibitor. Panel (**C**), distribution of predicted affinities of each identified peptide for the HLA alleles present in MOLT-4 cells *(HLA-A*01:01, HLA-A*25:01, HLA-B*18:01, HLA-B*57:01, HLA-C*06:02, HLA-C*12:03*) [28]. Each circle denotes a unique peptide sequence. Peptides are grouped as in Panel (**A**).

**Figure 4 ijms-23-01913-f004:**
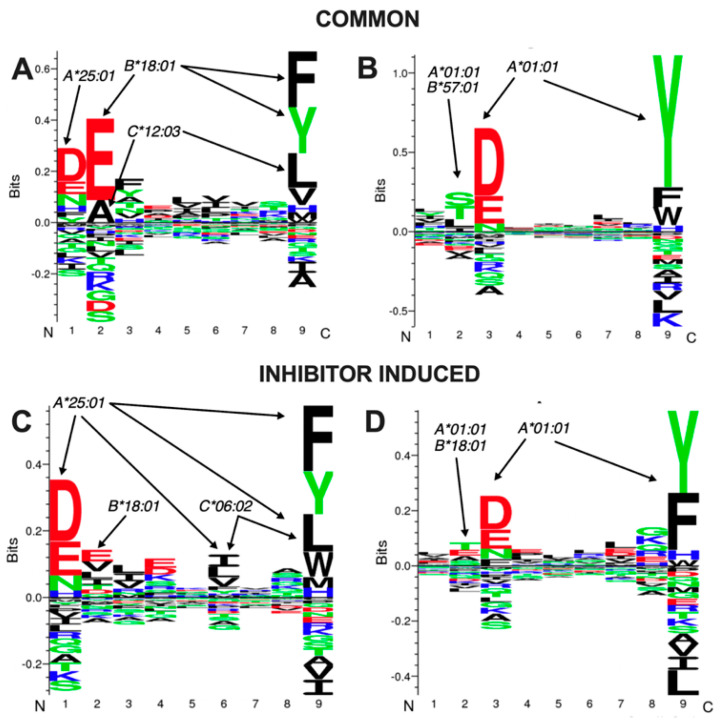
Weblogo type plots based on Gibbs cluster analysis of 9mer sequences of identified peptides. Analysis was performed using the GibbsCluster-2.0 Server and plotted using the Seq2logo server. Amino acids are colored based on their physicochemical properties (negatively charged = red, positively charged = blue, hydrophobic/aromatic = black, hydrophilic = green). The size of the letter representation of each amino acid single letter code indicates the probability of observation at the particular position of each cluster. Positive value on the y-axis suggests a higher-than-random prevalence of the particular residue at that position. Positions that show the enhanced presence of residues that correspond to anchor residues of particular HLA alleles are indicated with arrows. Panels (**A**,**B**) indicate the two major clusters observed for peptides common in both control and inhibitor conditions and panels (**C**,**D**) indicate the two major clusters observed for peptides unique in the inhibitor-treated sample.

**Figure 5 ijms-23-01913-f005:**
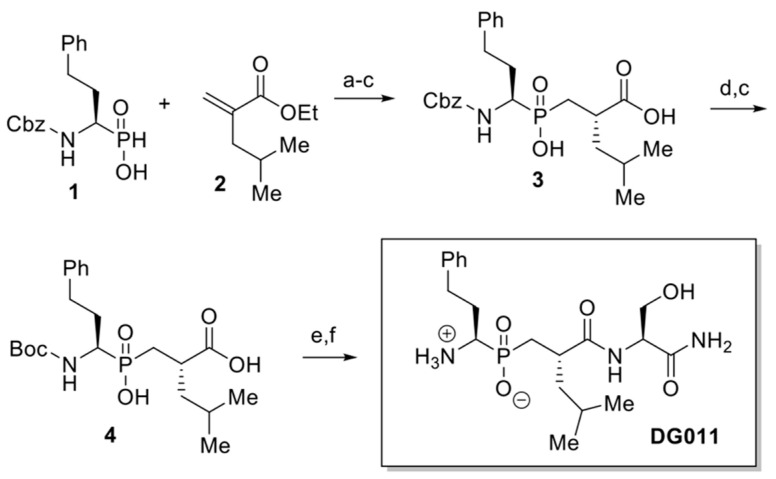
Synthetic strategy for DG011. (a) HMDS, 110 °C, 1 h, then 90 °C, 3 h, then EtOH, 70 °C, 30 min; (b) aq. NaOH, EtOH, rt, 24 h, then H_3_O+; (c) 2 × recrystallizations by AcOEt, 46%, three steps; (d) HBr/AcOH 33%, rt, 1 h; I Boc_2_O, Et_3_N, DMF, rt, 24 h, 92%, two steps; (e) H-(L)Ser(TBS)-NH_2_, EDC∙HCl, HOBt, DIPEA, CH_2_Cl_2_, rt, 4 h; (f) TFA/CH_2_Cl_2_/TIS/H_2_O 48:49:2:1, rt, 2 h, 43%, two steps.

## Data Availability

All data described are available in the article and associated Appendix A. Numerical values used for generation of graphs are available upon request to the corresponding author (Efstratios Stratikos; E-mail: stratos@rrp.demokritos.gr or estratikos@chem.uoa.gr). The MS proteomics raw data have been deposited to the ProteomeXchange Consortium via the PRIDE [32] partner repository with the dataset identifier PXD029895 (http://www.ebi.ac.uk/pride/archive/) (accessed on 27 December 2021).

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
