# Peer review of "ERAP2 Inhibition Induces Cell-Surface Presentation by MOLT-4 Leukemia Cancer Cells of Many Novel and Potentially Antigenic Peptides"

_ijms, 2022, doi:10.3390/ijms23031913_

Round 1
Reviewer 1 Report
Stratikos and coworkers a couple of years ago developed an inhibitor of ERAP2, a protease that is involved in the generation of antigenic peptides that are presented by MHC molecules on the cell surface and thereby and indirectly are involved in the modulation of the adaptive immune response. In this study the authors investigated the immunopeptidome of cancer cells that were treated with an ERAP2 inhibitor. To this end, the authors isolated MHC class I molecules and sequenced bound peptides by LC tandem MS. Interestingly, the authors found a shift of the immunopeptidome and identified peptides that were novel or upregulated. On a long run, these result may be helpful for evaluation of options for anti-ERAP2 cancer therapy and therefore merit publication. Moreover, the paper is clear and concise written. Typos that should be corrected:
Figure 1B: X-axis label: concetraiton should read concetration.
P 6 line 190 … is also likely remove italic formatting.
P6 li 193 … that ARE more sensitive…
Author Response
We thank the reviewer for his/her kind comments. All mentioned typos have now been corrected.
Reviewer 2 Report
The reviewed work presents very interesting results, although relating to a fairly limited research area. The authors made a good introduction to the topic of their research based on the literature data. This part well justifying high importance for science of their research. The presented research toolkit and the scope of the experimental work carried out are appropriate to the research hypothesis. The analytical methods used are sufficient to prove the main theses of this work. Although the authors did not distinguish a separate chapter containing the conclusions drawn from their research, they included a paragraph that starts with the words "In summary, we present ..." which may replace the chapter containing conclusions.
In my summary, I state that the evaluated work, within the scope of my competences, meets the criteria necessary to be accepted for publication in Int. J. Mol. Sci without any author's correction.
Author Response
We thank the reviewer for his/her kind comments. We have now expanded the concluding section of the manuscript (in part as a response to reviewer #3 comment) where we discuss additional points as well as limitations of this study.
Reviewer 3 Report
In the submitted study, “ERAP2 inhibition induces cell-surface presentation by cancer 2cells of many novel and potentially antigenic peptides”.
This study describes that DG011A can regulate immunopeptidome in MOLT4 cells and suggested that this could be an important tool in the fight against cancer.
Concerns:
- Authors have never mentioned the % of confluence of the culture cells used, despite of its importance to interpret data. Also, in addition to MTT other method to determine cell viability should be used.
- Authors need to test the DG011A in other leukaemia cancer cells or even in other type of cancer to determine if the effect claimed is conserved.
- It would be important to understand the effect of the coumpound in cancer cells with equivalent expression of ERAP1 and ERAP2.
IV) Despite MOLT4 present a higher expression of ERAP2 compared to ERAP1, authors need to silence ERAP1 to confirm the results claimed (otherwise they cannot exclude the contribution of ERAP1). V) In results and discussion section, authors should be more elucidative explaining the impact and meaning of data obtained. From a reader point of view interpretation of graphs and table should be done to clarify the authors’ message and scientific achievements.VI) Integration of data in the literature is crucial. Also an explanation or possible explanations (mechanistic or other) should be incorporated.
Author Response
Comment #1
Authors have never mentioned the % of confluence of the culture cells used, despite of its importance to interpret data. Also, in addition to MTT other method to determine cell viability should be used.
Response: Although grown in T175 tissue culture flasks MOLT4 cells are mostly in suspension in the culture medium, so confluency cannot be determined. Instead, cells are counted before each passage, to maintain a density between 4x105-2x106 cells/mL. This is now mentioned in the methods section. Cell viability was also routinely followed by Trypan Blue during passaging and was >95% at all stages. The presence of the inhibitor did not affect cell viability or growth rates during culture.
Comment #2
Authors need to test the DG011A in other leukaemia cancer cells or even in other type of cancer to determine if the effect claimed is conserved.
Response: This is a very solid idea worthwhile exploring. We do not claim that this effect is conserved in all leukemia cell lines and we have no reason to believe that it does. Indeed, it has been previously shown that the relative expression levels of ERAP1 and ERAP2 vary asynchronously in cancer cell lines (J Cell Physiol. 2008 Sep;216(3):742-9.) thus the level of influence of ERAP2 on the cellular immunopeptidome may vary also. Our intent was to provide a proof-of-concept study that would evaluate the effect of chemical inhibition of ERAP2 in a native cell culture system and we intentionally selected a cell line that has good ERAP2 expression to increase the likelihood of observing effects. ERAP2 inhibition in low-ERAP2 expression or high-ERAP1 expression cell lines may not lead to similar results, thus limiting therapeutic avenues only to cancers that over-express ERAP2. This is now discussed at the end of the manuscript. Regardless, comparative testing other cancer cell lines is potentially useful but cannot be performed in the context of this study since it would require the repetition of all experiments so that proper proteomic comparisons can be performed.
Comment #3
It would be important to understand the effect of the coumpound in cancer cells with equivalent expression of ERAP1 and ERAP2.
Response: This is a valid idea, that however is largely covered by our response to comment #2. Testing the importance of ERAP2 on the immunopeptidome in different cancer cell lines with varied relative expression of ERAP1 and ERAP2 would be valuable but is outside the scope of this proof-of-concept study and cannot be performed in a timely manner because comparing proteomics data necessitates parallel experiments and would require the repetition of all experiments presented here.
Comment #4
Despite MOLT4 present a higher expression of ERAP2 compared to ERAP1, authors need to silence ERAP1 to confirm the results claimed (otherwise they cannot exclude the contribution of ERAP1).
Response: We intentionally selected a cell line that has high levels of expression of ERAP2 versus ERAP1, to help us discern ERAP2-mediated effects and to simulate a potential clinical system in which ERAP2 is a worthwhile target. However, we feel that the presence of ERAP1 is advantageous since it allows us to discern effects of ERAP2 inhibition in the more “native” state of ERAP1 co-expression. It is likely that the low levels of ERAP1 still have some contribution to shaping the immunopeptidome of this cell line and we wanted to see if manipulation of ERAP2 activity by the inhibitor, in the presence of ERAP1, leads to changes in the immunopeptidome. While knocking out ERAP1 would make the system better controlled, we feel that it would take away from the project since it should simulate more rare occasions when ERAP1 is not expressed at all. The rational for selecting this cell line are now better explained in the introduction section.
Comment #5
In results and discussion section, authors should be more elucidative explaining the impact and meaning of data obtained. From a reader point of view interpretation of graphs and table should be done to clarify the authors’ message and scientific achievements.
Response: We have added additional text towards the end of the manuscript in which we further elucidate our findings and interpret them in both our understanding of ERAP2 biology and pharmacology. In addition, we discuss limitations of this study as well as future directions.
Comment #6
Integration of data in the literature is crucial. Also an explanation or possible explanations (mechanistic or other) should be incorporated.
Response: We are not clear on what the reviewer means with integration of data in the literature. We have added an additional text discussing the results in which we better address the mechanistic explanations of the observations.
Round 2
Reviewer 3 Report
I beleive you have addressed my concerns regarding most points. I only think that the work could bring more informaton to the field if other cancer cell lines with different expression profiles for ERAP2 would be tested.
Author Response
We agree with the reviewer that expanding this kind of analysis to additional cell lines with different levels of ERAP2 expression would give valuable information. However, currently it is not readily possible to apply this approach on several cell lines for practical reasons. The proteomic experiment described in this manuscript requires large-scale cell culture (10^9 cells) and purification which necessitates about 1 year of work (making 20-30mg of monoclonal antibody, columns, growing the cells, synthesizing sufficient amounts of the inhibitor, time consuming and expensive LC-MS/MS analysis) for a single cell line. It is not financially feasible for our lab to perform several such rounds to address this concern. It is more likely that another, higher-throughput approach, (i.e. looking at T cell responses) would be more appropriate to explore the effects of ERAP2 inhibition in the antigenicity of cancer cells but we don't currently have the tools to perform this experiment.